# Non-Surgical Camouflage Treatment of a Skeletal Class III Patient with Anterior Open Bite and Asymmetry Using Orthodontic Miniscrews and Intermaxillary Elastics

**Ye-Ji Seo** [1,†], **Jae Hyun Park** [2,3,†], **Na-Young Chang** [1,4] and **Jong-Moon Chae** [1,2,4,*]

1   Department of Orthodontics, School of Dentistry, University of Wonkwang, Iksan 54538, Republic of Korea
2   Postgraduate Orthodontic Program, Arizona School of Dentistry & Oral Health, A.T. Still University, Mesa, AZ 85206, USA
3   Graduate School of Dentistry, Kyung Hee University, Seoul 02453, Republic of Korea
4   Wonkwang Dental Research Institute, University of Wonkwang, Iksan 54538, Republic of Korea
*   Correspondence: jongmoon@wku.ac.kr
†   These authors contributed equally to this work.

**Abstract:** This case report presents the non-surgical orthodontic camouflage treatment of an 18-year-old male patient with skeletal Class III asymmetry and severe anterior open bite. The anterior open bite was corrected by extrusion of the maxillary and mandibular anterior teeth and clockwise and counterclockwise rotation of the maxillary and mandibular occlusal plane, respectively, using intermaxillary Class III elastics between the maxillary posterior buccal miniscrews and mandibular canines and anterior vertical elastics between the maxillary and mandibular canines. Class III dental relationships and dental asymmetry were corrected via unilateral distalization of the mandibular dentition on the left side using a closed coil spring between the buccal shelf screw and hook. The patient's smile esthetics and dental relationship were improved with a more favorable facial profile, and facial asymmetry was slightly alleviated after orthodontic camouflage treatment. The total treatment time was 15 months. A modified wraparound retainer with a scalloped labial bow, tongue crib, and tooth positioner was used simultaneously to prevent the potential relapse.

**Keywords:** skeletal class III malocclusion; anterior open bite; asymmetry; miniscrew; non-surgical orthodontic camouflage treatment

## 1. Introduction

The combination of skeletal Class III malocclusion, anterior open bite, and asymmetry has been considered one of the most difficult problems to treat in orthodontics. In adult patients with these problems, orthognathic surgery with orthodontic decompensation [1–6] or non-surgical orthodontic camouflage treatment [7–24] would be the appropriate treatment options.

Non-surgical correction of skeletal Class III malocclusion can be performed by total distalization of the mandibular dentition, protraction of the maxillary dentition, and clockwise rotation of the mandible [9,12–14,18,19]. Non-surgical correction of an anterior open bite can be accomplished by the intrusion of the posterior teeth, extrusion of the anterior teeth, counterclockwise autorotation of the mandible, and their combination [7–11,16,17,20,21,25,26]. Non-surgical correction of mandibular dental asymmetry can be achieved by unilateral retraction of the mandibular dentition with skeletal anchorage, but skeletal asymmetry can only be corrected to a limited extent [19,20].

The intrusion of the posterior teeth induces a counterclockwise autorotation of the mandible, which can improve the open bite but can also degrade a patient's profile to a more concave pattern in a Class III patient. Therefore, orthodontists should consider changes in the anteroposterior relationship as a result of vertical movement of dentition when treating cases where there is a combination of open bite and Class III malocclusion [9,15,20,22–24].

The dental midline, amount of exposure at rest, and inclination of the maxillary central incisors should be carefully examined to achieve an acceptable camouflage treatment result. Therefore, the three-dimensional position of the maxillary incisors should be considered as a starting point in determining the pattern of orthodontic tooth movement to achieve an acceptable facial appearance and dental occlusion [27].

The present case report describes the non-surgical orthodontic camouflage treatment of a young adult patient with skeletal Class III malocclusion, anterior open bite, and asymmetry using skeletal anchorage and intermaxillary elastics.

## 2. Case Report

### 2.1. Diagnosis and Etiology

An 18-year-old male patient presented with the chief complaints of anterior open bite and chin protrusion. He was in good health with no medical or dental history. Facially, he showed a concave profile due to a protruded chin. And he exhibited a chin deviation toward the right side compared to the facial midline with a slight right-up lip canting. When smiling, he displayed no incisor display and looked older than his age. Intraorally, the mandibular dental midline deviated about 4.2 mm to the right compared to the maxillary dental midline. He also showed a severe anterior open bite extended to the first premolars, Class III canine and molar relationships with no crowding or spacing, −2.1 mm of overjet, and −7.1 mm of overbite. The required amount of distalization at the crown level on the dental casts was 3.0 mm and 4.0 mm on the right and left sides to establish Class I molar relationships, respectively (Figures 1 and 2).

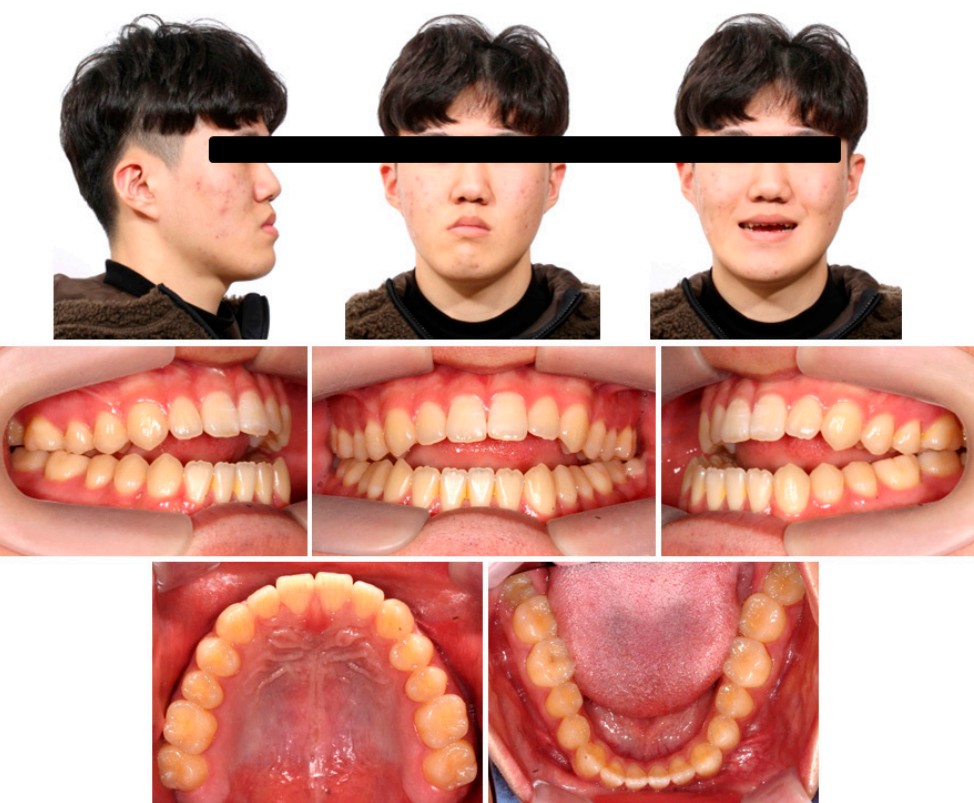

**Figure 1.** Pretreatment facial and intraoral photographs.

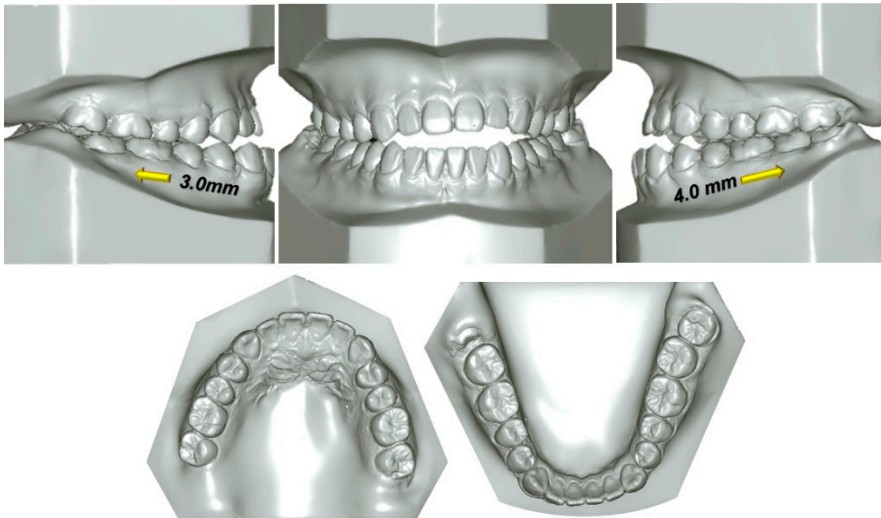

**Figure 2.** Pretreatment dental casts. The required amount of distalization at the crown level on the dental cast (Right, 3.0 mm; Left, 4.0 mm) to establish a Class I molar relationship.

The panoramic radiograph showed the presence of four third molars. The lateral cephalometric analysis indicated a skeletal Class III relationship with a protruded mandible (SNA, 81.7°; SNB, 85.4°; ANB, −3.7°; AO-BO, −9.8 mm; APDI, 92.4), and a normal vertical facial pattern [mandibular plane angle (SN-MP), 29.5°; facial height ratio, 70.5%]. The maxillary and mandibular incisors were compensated (U1 to FH, 127.6°; IMPA, 86.7°) by a skeletal Class III pattern. The maxillary occlusal plane angle was −3.5° and the mandibular occlusal plane angle was 7.8°. The upper and lower lips to the esthetic line were −4.8 mm and −1.3 mm, respectively. NLA was 99.5°, and the Z angle was 85.7° (Figure 3, Table 1).

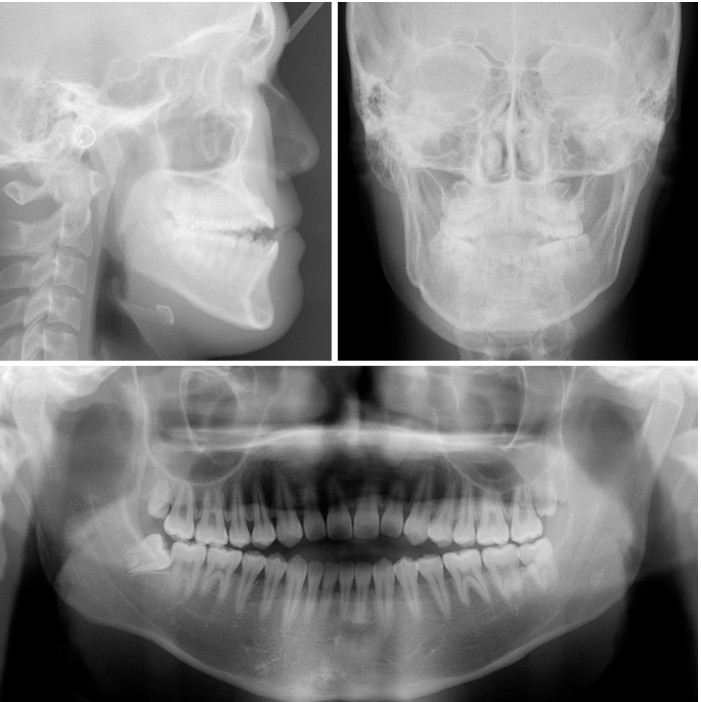

**Figure 3.** Pretreatment radiographs.

**Table 1.** Two-dimensional cephalometric measurements.

| Variables | Norm | T1 | T2 (6 Months) | T3 (15 Months) | T4 (22 Months) |
|---|---|---|---|---|---|
| SNA (°) | 82.4 | 81.7 | 81.7 | 81.4 | 81.4 |
| SNB (°) | 80.4 | 85.4 | 85.4 | 84.6 | 84.6 |
| ANB (°) | 2.0 | −3.7 | −3.7 | −3.2 | −3.2 |
| AO-BO (mm) | −2.2 | −9.8 | −8.5 | −6.6 | −5.2 |
| APDI | 85.9 | 92.4 | 92.3 | 91.4 | 90.8 |
| ODI | 73.3 | 55.8 | 55.2 | 55.5 | 55.1 |
| SN-MP (°) | 30.2 | 29.5 | 30.0 | 31.2 | 31.2 |
| AFH (mm) | 136.4 | 140.8 | 141.1 | 143.0 | 142.8 |
| PFH (mm) | 95.4 | 99.2 | 98.5 | 99.2 | 98.9 |
| FHR (%) | 70.2 | 70.5 | 69.8 | 69.4 | 69.3 |
| U1 to FH (°) | 116.0 | 127.6 | 121.8 | 126.5 | 127.5 |
| IMPA (°) | 96.6 | 86.7 | 82.1 | 72.5 | 76.2 |
| U1 to L1 (°) | 124.0 | 123.0 | 132.8 | 136.7 | 133.0 |
| MxOP to FH (°) | 14.0 | −3.5 | 1.5 | 2.2 | 1.1 |
| MnOP to FH (°) | 14.0 | 7.8 | −0.9 | −0.9 | 0.3 |
| UL to EL (mm) | 1.0 | −4.8 | −2.4 | −2.8 | −2.7 |
| LL to EL (mm) | 1.0 | −1.3 | 1.4 | 0.3 | −1.1 |
| NLA (°) | 100.0 | 99.5 | 91.6 | 92.7 | 92.8 |
| Z angle (°) | 75.0 | 85.7 | 81.0 | 83.1 | 85.9 |

T1, pretreatment; T2, midtreatment (6 months); T3, posttreatment (15 months); T4, retention (22 months). SNA, angle between sella-nasion (SN) and nasion-point A (NA); SNB, angle between SN and nasion-point B (NB); ANB, difference between the SNA and SNB angles; AO-BO, distance between perpendiculars drawn from point A and point B onto the occlusal plane; APDI, anteroposterior dysplasia indicator; ODI, overbite depth indicator; SN-MP, angle between sella-nasion plane and mandibular plane; AFH (anterior facial height), linear measurement from nasion to menton; PFH (posterior facial height), linear measurement from sella to gonion; FHR (facial height ratio), ratio of PFH to AFH; FH plane, Frankfort horizontal plane; UI to FH, angle between maxillary incisal axis and FH plane; IMPA, angle between mandibular incisal axis and mandibular plane; U1 to L1, angle between maxillary and mandibular central incisal axes; OP to FH, angle between occlusal plane to FH; Mx, maxilla; Mn, mandible; UL to EL, distance between upper lip to esthetic line; LL to EL, distance between lower lip to esthetic line; NLA, nasolabial angle; Z angle, angle between FH and profile line tangent to chin and prominent vermilion border of both lips.

Cone-beam computed tomography (CBCT) images showed dental deviation to the right side of 2.0 mm and 6.2 mm in the maxilla and mandible, respectively, and 7.7 mm chin (at menton) deviation to the right side due to mandibular yawing with a 0.8 mm difference in the mandibular body length between right and left sides. CBCT images were reoriented, and measurements were made using ON3D (3D ONS, INC., Seoul, South Korea) software. The distance between the maxillary central incisal tip and stomion was deficient, causing no incisor display when smiling. The 3D coordinates (x, y, z) were constructed using nasion (N) as the reference point (0, 0, 0) (Figure 4, Table 2). Posterior available spaces were 3.9 mm and 4.0 mm at root apex level (Figure 5A) and 6.5 mm and 6.0 mm at crown level (Figure 5B) on the right and left sides, respectively.

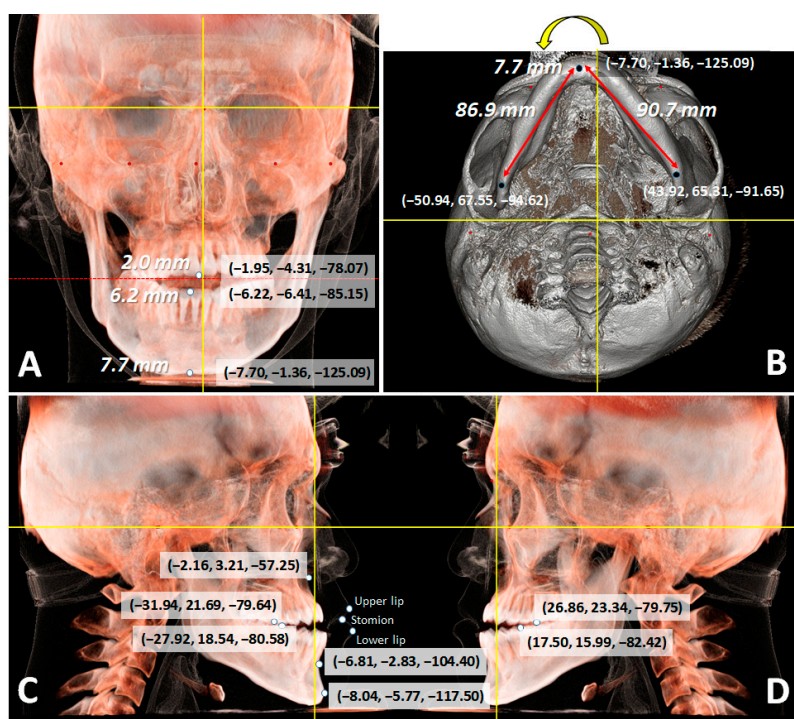

**Figure 4.** Pretreatment CBCT images using 3-dimensional coordinates. (**A**) Frontal image; (**B**) Submentovertex image (arrow means mandibular yawing); (**C**) Sagittal image (right side); (**D**) Sagittal image (left side).

**Table 2.** Three-dimensional cephalometric measurements.

| Variables | T1 (x, y, z) | T3 (x, y, z) |
|:---:|:---:|:---:|
| U1MP | $(-1.95, -4.31, -78.07)$ | $(-2.04, -3.66, -81.29)$ |
| L1MP | $(-6.22, -6.41, -85.15)$ | $(-1.82, -0.90, -79.89)$ |
| RU6CP (right) | $(-31.94, 21.69, -79.64)$ | $(-31.43, 22.41, -80.25)$ |
| LU6CP (left) | $(26.86, 23.34, -79.75)$ | $(25.86, 23.42, -81.27)$ |
| RL6CP (right) | $(-27.92, 18.54, -80.58)$ | $(-28.35, 20.14, -79.59)$ |
| LL6CP (left) | $(17.50, 15.99, -82.42)$ | $(22.63, 21.56, -79.98)$ |
| Point A | $(-2.16, 3.21, -57.25)$ | $(-1.03, 3.25, -58.82)$ |
| Point B | $(-6.81, -2.83, -104.40)$ | $(-4.98, -1.87, -103.55)$ |
| Pogonion | $(-8.04, -5.77, -117.50)$ | $(-6.13, -5.20, -119.91)$ |
| Menton | $(-7.70, -1.36, -125.09)$ | $(-5.46, 0.09, -126.68)$ |
| Gonion (right) | $(-50.94, 67.55, -94.62)$ | $(-49.93, 69.01, -94.76)$ |
| Gonion (left) | $(43.92, 65.31, -91.65)$ | $(44.47, 65.37, -93.14)$ |
| Stomion | $(-2.69, -14.54, -77.29)$ | $(-1.37, -14.40, -80.00)$ |
| UL | $(-2.05, -19.23, -71.22)$ | $(-0.84, -19.52, -72.61)$ |
| LL | $(-3.41, -22.19, -84.05)$ | $(-0.64, -20.06, -87.60)$ |

T1, pretreatment; T3, posttreatment (15 months); U1MP, the midpoint of maxillary central incisal tips; L1MP, the midpoint of mandibular central incisal tips; R, right; L, left; U6CP, mesiobuccal cuspal point of maxillary first molar; L6CP, mesiobuccal cuspal point of mandibular first molar; UL, most anterior point of upper lip; LL, most anterior point of the lower lip.

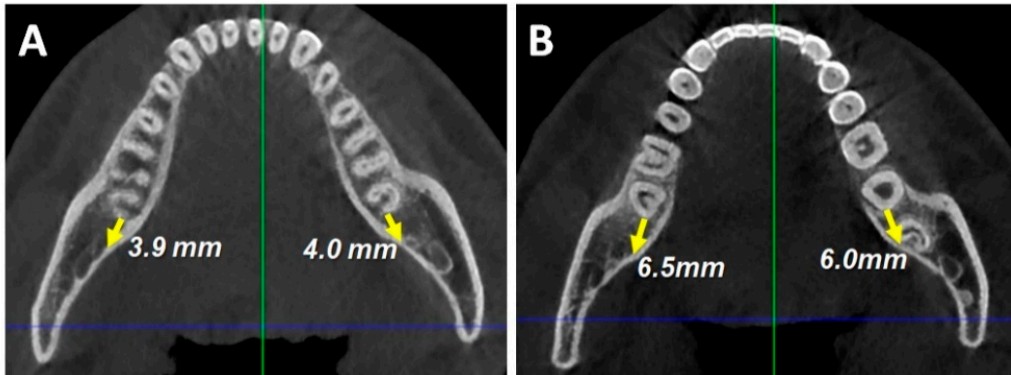

**Figure 5.** Posterior available spaces for distalization of the mandibular posterior teeth on CBCT axial images. (**A**) At root apex level. (**B**) At cementoenamel junction (CEJ) level.

### 2.2. Treatment Objectives

The treatment objectives were to (1) achieve a proper anterior overbite and overjet, (2) obtain Class I canine and molar relationships, (3) correct the dental midline deviation, (4) achieve a stable occlusal relationship, and (5) improve the facial profile.

### 2.3. Treatment Plan

The ideal treatment plan was a combination of surgical and orthodontic treatment. Two-jaw surgery with genioplasty was planned to correct the skeletal Class III asymmetry and open bite. But the patient and his parents refused the orthognathic surgery, so a different treatment plan was needed.

The alternative treatment plan was non-surgical orthodontic camouflage treatment using miniscrews after extraction of the third molars, followed by extrusion of the incisors and total distalization of the mandibular dentition to achieve acceptable treatment results. The authors fully explained the advantages (less aggressive procedure, low cost, acceptable treatment results without orthognathic surgery) and disadvantages (high relapse tendency, the possibility of miniscrew failure, compromised treatment results) of this non-surgical procedure to the patient. The patient chose this treatment option. He understood the limitations (possibility of relapse, compromise of facial, skeletal, and dental correction, and the difficulty of improving facial and skeletal asymmetry) of orthodontic camouflage treatment without orthognathic surgery.

### 2.4. Treatment Progress

After extraction of the third molars, the transpalatal arch was delivered on the maxillary first molars, and 0.022 × 0.028-in edgewise appliance were placed in both arches, and leveling began with a 0.014-in nickel-titanium (NiTi) and 0.018-in and 0.018 × 0.025-in stainless steel archwires. Two maxillary interradicular miniscrews (1.6 mm in diameter, 8 mm in length; Jeil Medical, Seoul, Republic of Korea) were placed between the maxillary second premolars and first molars. Class III intermaxillary elastics were applied between the maxillary posterior miniscrews and the mandibular canines, and vertical elastics were applied between the maxillary and mandibular canines. Six months into treatment, the anterior open bite was improved by extrusion of the incisors and rotation of the occlusal planes (Figure 6).

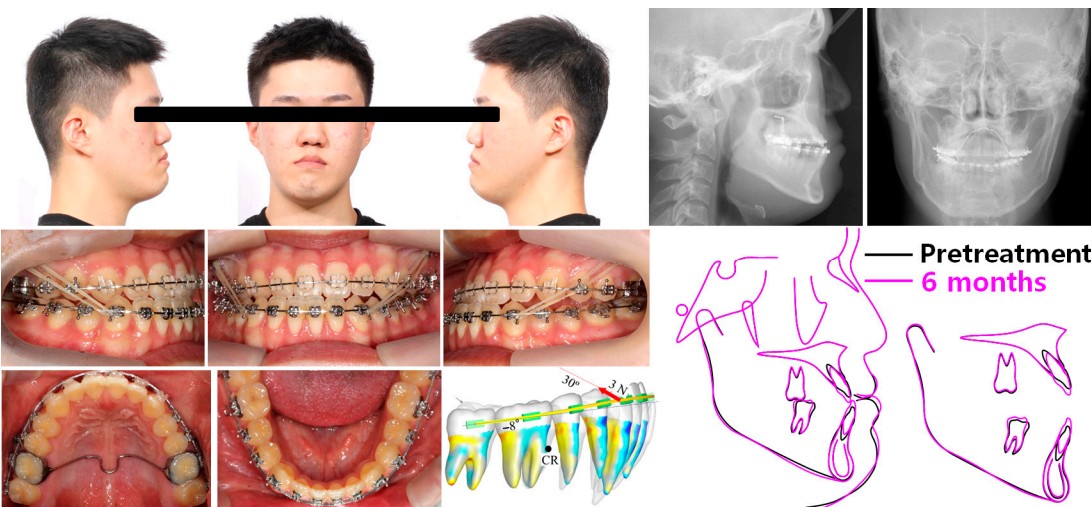

**Figure 6.** Treatment progress (6 months) photographs, schematic diagram of biomechanics, and cephalometric radiographs and superimposition.

At eight months into treatment, a mandibular miniscrew was placed on the mandibular left buccal shelf area between the first and second molars. A closed coil spring was applied between the anterior hook and buccal shelf screw to correct the mandibular dental midline deviation and the Class III relationship by unilateral distalization of the mandibular dentition on the left side (Figure 7A). At eleven months of treatment, the dental midline was corrected, and positive overbite and overjet were obtained. Class III and vertical intermaxillary elastics were additionally applied to improve the occlusion (Figure 7B). At thirteen months of treatment, a closed coil spring in the anterior maxillary teeth and elastomeric chains in the mandibular arch were applied for space closure and detailing (Figure 7C).

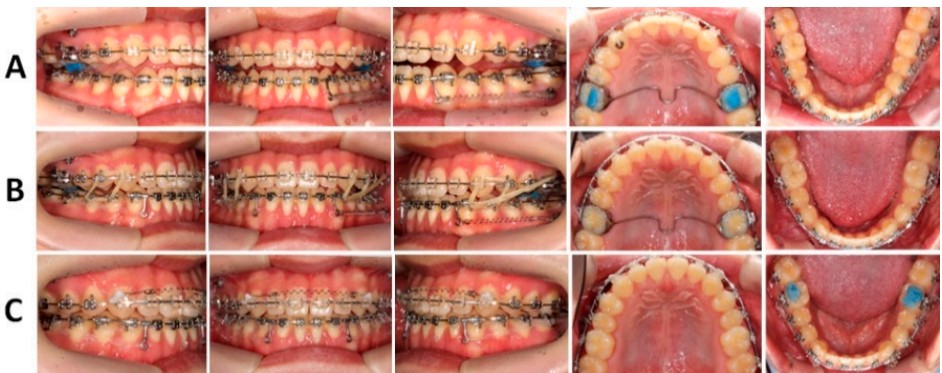

**Figure 7.** Treatment progress intraoral photographs ((**A**) 8 months; (**B**) 11 months; (**C**) 13 months).

At fifteen months into treatment, fixed appliances were removed. Maxillary and mandibular anterior teeth were stabilized with canine-to-canine fixed lingual retainers, and wraparound retainers were delivered in both arches. A scalloped labial bow along the cementoenamel junction of the anterior maxillary teeth with a tongue crib was additionally placed in the maxillary retainer to prevent a relapse of the open bite. Also, a positioner was delivered to improve the anteroposterior and vertical dental relationships. The patient was instructed to wear the wraparound-type retainers during the daytime and the positioner at night (Figure 8).

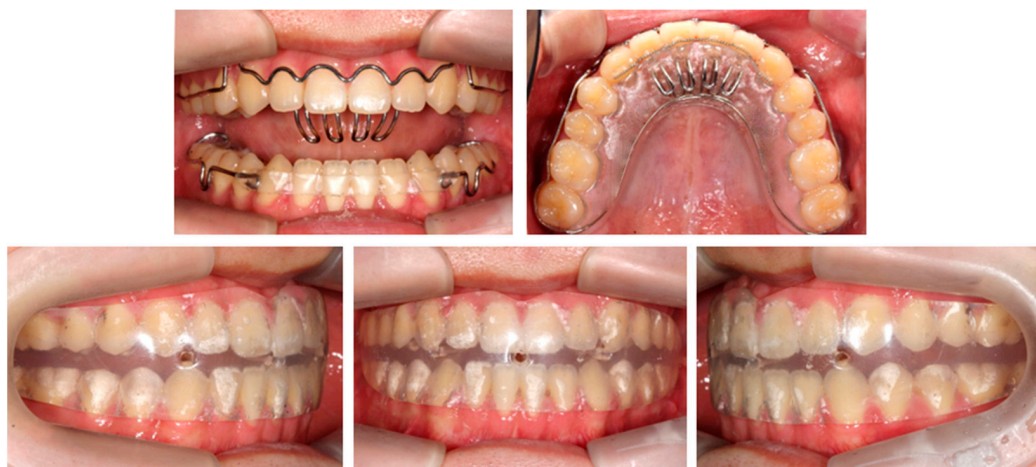

**Figure 8.** Retainers. **Top**, wraparound-type retainers; **Bottom**, tooth positioner.

*2.5. Treatment Results*

The posttreatment facial photographs and cephalograms showed an increased exposure of the anterior maxillary teeth when smiling with their extrusion and an acceptable facial profile and symmetry. Posttreatment intraoral photographs and dental casts showed a Class I canine and molar relationships with good occlusal interdigitation, maxillary and mandibular dental midline coincidence, and acceptable overjet and overbite. The posttreatment panoramic radiograph showed proper root parallelism. The CBCT axial image at the root apex level showed that the roots of the mandibular posterior teeth were in touch or penetrated into the lingual cortical bone by distalization of the mandibular posterior teeth (Figures 9–11).

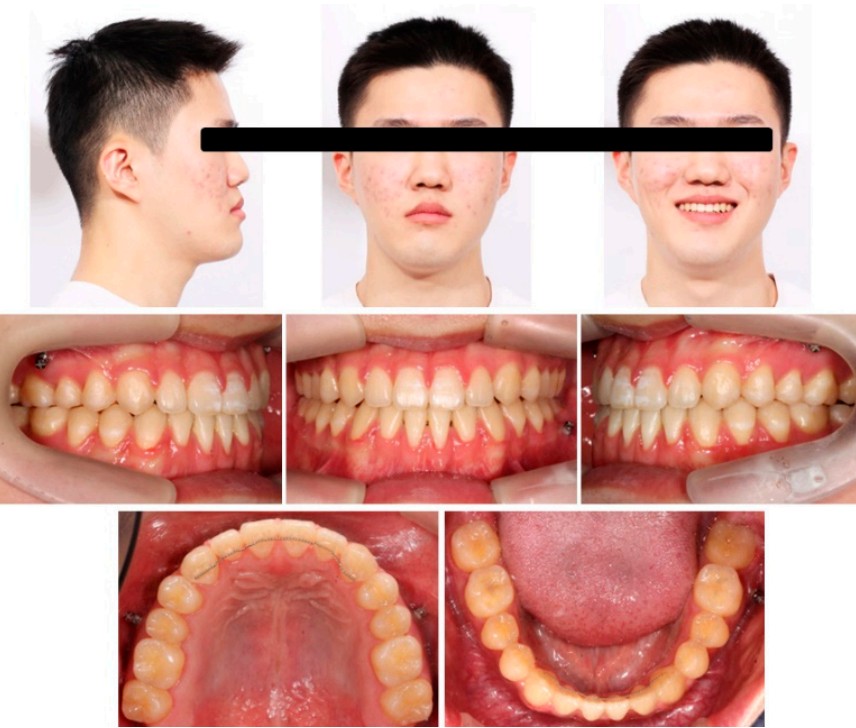

**Figure 9.** Posttreatment facial and intraoral photographs.

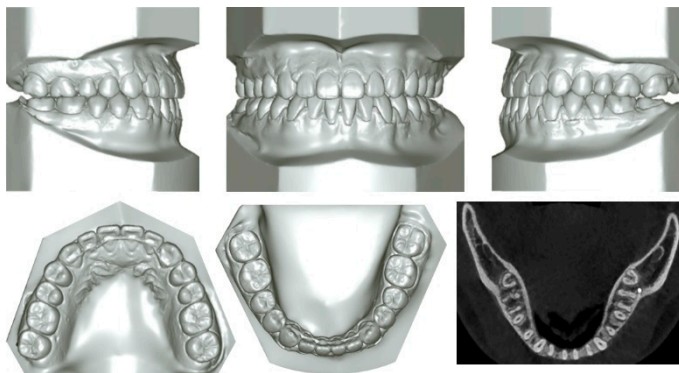

**Figure 10.** Posttreatment dental casts and CBCT axial image of the mandible.

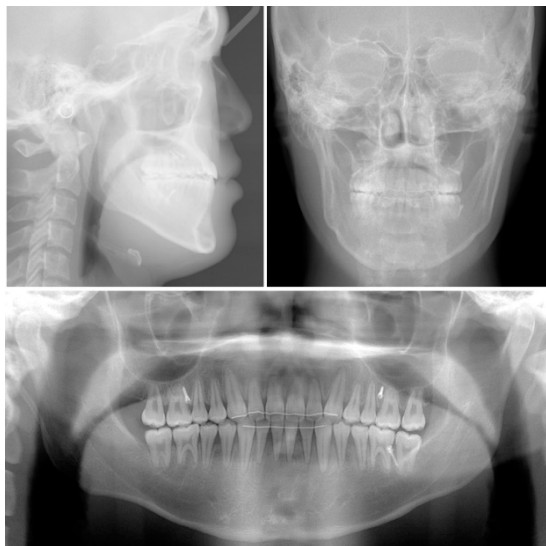

**Figure 11.** Posttreatment radiographs.

Cephalometric superimpositions revealed that the anterior maxillary teeth were extruded with clockwise rotation of the maxillary occlusal plane, and the mandibular anterior and posterior teeth were extruded and uprighted with counterclockwise rotation of the mandibular occlusal plane. And the mandible rotated downward and backward with increasing anterior facial height (AFH) followed by improvement of the facial profile (Figure 12).

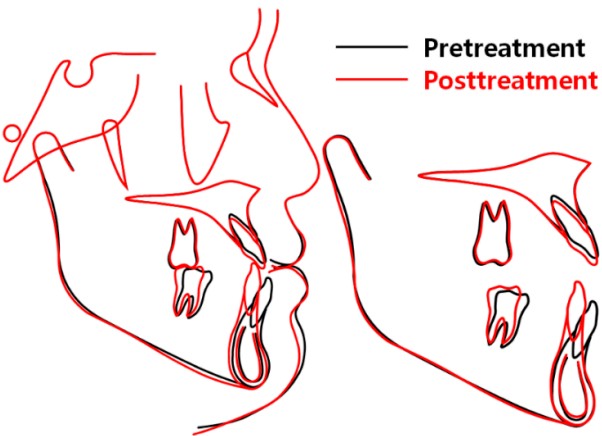

**Figure 12.** Cephalometric superimposition. Black, pretreatment; Red, posttreatment (15 months).

The ANB angle increased by 0.5° from −3.7° to −3.2°, the Wits value increased by 3.2 mm from −9.8 mm to −6.6 mm, the mandibular plane angle (SN-MP) increased by 1.7° from 29.5° to 31.2°, facial height ratio decreased by 1.1% from 70.5% to 69.4%, U1 to FH decreased by 1.1° from 127.6° to 126.5°, IMPA decreased by 14.2° from 86.7° to 72.5°, interincisal angle increased by 13.7° from 123.0° to 136.7°, maxillary occlusal plane angle increased by 5.7° from −3.5° to 2.2°, mandibular occlusal plane angle decreased by 8.7° from 7.8° to −0.9°, UL-EL increased by 2.0 mm from −4.8 mm to −2.8 mm, LL-EL increased by 1.6 mm from −1.3 mm to 0.3 mm, nasolabial angle decreased by 6.8° from 99.5° to 92.7°, and Z angle decreased by 2.6° from 85.7° to 83.1° (Table 1).

Posttreatment CBCT images showed that U1MP moved inferiorly by 3.2 mm from 78.1 mm to 81.3 mm and L1MP moved superiorly by 5.3 mm from 85.2 mm to 79.9 mm. Mandibular dental deviation was reduced by 4.4 mm from 6.2 mm to 1.8 mm, but the maxillary dental midline showed little change. Chin deviation (menton) was reduced by 2.2 mm from 7.7 mm to 5.5 mm. The pogonion, menton, stomion, upper lip, and lower lip moved downward (Figure 13, Table 2). At 22 months into retention, the treatment results were well maintained except for a slight relapse in the mandibular dental midline, anterior teeth, and lips (Figures 14 and 15).

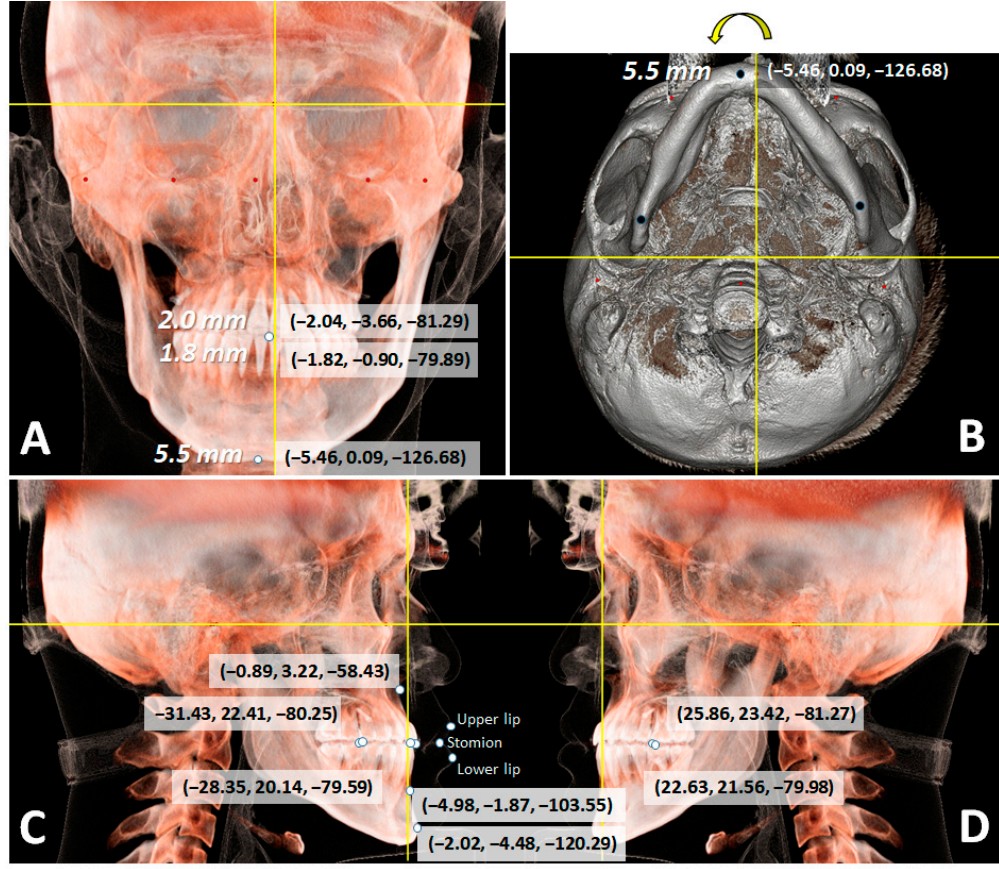

**Figure 13.** Posttreatment CBCT images using 3-dimensional coordinates. (**A**) Frontal image; (**B**) Submentovertex image (arrow means mandibular yawing); (**C**) Sagittal image (right side); (**D**) Sagittal image (left side).

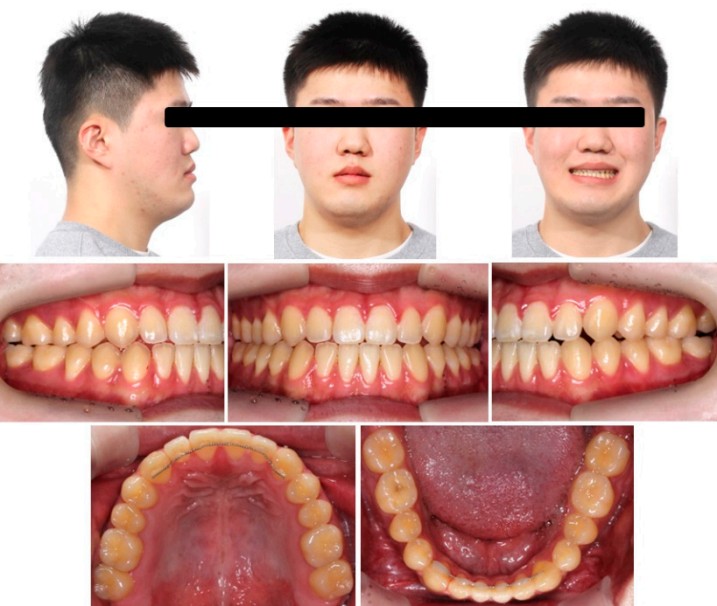

**Figure 14.** 22-month retention facial and intraoral photographs.

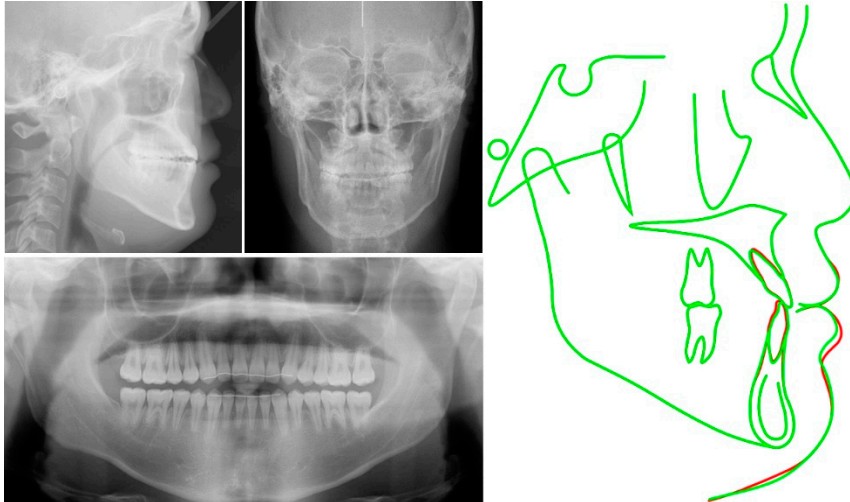

**Figure 15.** 22-month retention radiographs and cephalometric superimposition. Red, posttreatment; Green, 22-month retention.

## 3. Discussion

Orthognathic surgery combined with orthodontic treatment has been well known as a clinically safe and stable procedure for the correction of skeletal Class III, open bite, and asymmetry [1–6]. However, with the advent of skeletal anchorage, non-surgical approaches have become more common for the correction of skeletal malocclusions using miniscrews or miniplates [7–24].

The patient, in this case, showed an anterior open bite with a Class III skeletal pattern and insufficient maxillary incisor exposure when smiling. Kim [17] suggested that the vertical cephalometric position of the maxillary central incisors relative to the lip line should be about 4 mm downward and a guide for the anterior limit of the maxillary occlusal plane, and the vertical position of the mandibular incisors should be positioned at the level of stomion in relation to that of the maxillary incisor position to obtain a proper overbite. Woo et al. [28] and Kim et al. [29] also reported that the vertical distances between the maxillary incisal tip and the stomion were 2.7 mm and 2.1 mm, respectively.

In this patient, U1MP and L1MP were positioned 0.8 mm and 7.9 mm inferior to the stomion, respectively, at the lip, closed position, and no incisor display was shown when smiling. Therefore, extrusion of the incisors in both arches and control of the occlusal plane [25] was necessary to close the open bite and improve the smile aesthetics. The intrusion of the posterior teeth followed by autorotation of the mandible might be necessary to correct an open bite with a Class II skeletal pattern. But in this skeletal Class III patient, some extrusion of the posterior teeth was necessary to improve the facial profile. Backward rotation of the mandible was accomplished by increasing AFH. But this camouflage treatment worsened the vertical proportion of the face because the patient initially had a normal AFH. This might be a drawback with the non-surgical camouflage treatment of Class III malocclusion with anterior open bite.

At 6 months into treatment, the vertical position of the stomion had moved slightly anterosuperiorly even though AFH was slightly increased due to uprighting and extrusion of the mandibular molars. This might have been due to the extrusion of the mandibular incisors. At posttreatment, the stomion had moved posteroinferiorly due to an increase in AFH and retraction of the mandibular incisors, which means that more extrusion of the maxillary incisors was still necessary to obtain the proper smile esthetics. In our patient, the vertical position of his maxillary incisal tips was not appropriate in relation to the lip line due to the downward movement of the stomion. Fortunately, he showed an acceptable exposure of the maxillary incisors when smiling posttreatment because he had a hyperactive muscle that elevated the upper lip. Unfortunately, the distance between the stomion and maxillary incisal tip will steadily decrease throughout his life due to the continuous elongation of his upper lip [30].

The patient showed a skeletal Class III malocclusion with open bite due to the downward position of his mandibular incisors, so extrusion and uprighting of his mandibular incisors and molars with a counterclockwise rotation of the mandibular occlusal plane and total distalization of the mandibular dentition were necessary to correct the open bite and Class III dental relationship. In dental casts, Class III molar relationships can be corrected into Class I molar relationships by distalization of the mandibular molars by 3.0 mm and 4.0 mm on the right and left sides (Figure 2). Posterior available spaces were sufficient for the distalization of the mandibular molars after extraction of the third molars (Figure 5).

Chae et al. [12] conducted a biomechanical analysis for total distalization of the mandibular dentition using finite element analysis. They suggested that the selective use of force angulation might be helpful in achieving the proper tooth movement in each case, depending on the type of malocclusion. With our patient, Class III intermaxillary elastics were used with maxillary posterior miniscrews to achieve counterclockwise rotation of the mandibular occlusal plane and distalization of the whole mandibular dentition using a force angulation of 30° (Figure 6). As a result, slight extrusion of the mandibular first molars and moderate extrusion of the mandibular incisors were obtained without extrusion of the posterior maxillary teeth, even though Class III elastics were applied. On the other hand, conventional Class III elastics could have been used to extrude the posterior maxillary teeth increasing the AFH and intruding the anterior maxillary teeth reducing the amount of incisor exposure. They might cause a severe increase in the AFH and deterioration of smile esthetics. Therefore, applying selective biomechanics using skeletal anchorage is recommended to achieve acceptable camouflage treatment results.

Possible relapse is one factor that makes Class III malocclusion and anterior open bite treatment difficult. An open bite can be corrected non-surgically by the intrusion of the posterior teeth, extrusion of the anterior teeth, or a combination thereof. Deguchi et al. [31] indicated a relapse tendency in the group of incisor extrusion than in the molar intrusion group. Our patient needed to have his incisors extruded rather than a molar intrusion, so reinforcement of the retention was essential. Also, in this patient, skeletal Class III malocclusion was corrected by total distalization of the mandibular dentition. Song et al. [32] concluded that there was some relapse after distalization of the dentition, and Ning and Duan [33] suggested that a forceful tongue might cause a relapse. So, the authors

instructed the patient to wear a modified wraparound retainer with a tongue crib [34] and positioner [35] to reduce the chance of relapse of the extruded anterior teeth and distalized mandibular dentition.

Skeletal and dental asymmetries were seen in this patient. Skeletal asymmetry could not be corrected, but the patient understood the limitations of the camouflage treatment. Dental asymmetry was corrected by unilateral distalization of the mandibular dentition using skeletal anchorage. After treatment, mandibular asymmetry was slightly reduced, which might be due to an inherent functional asymmetry with a differential condylar space (Figure 16) [36].

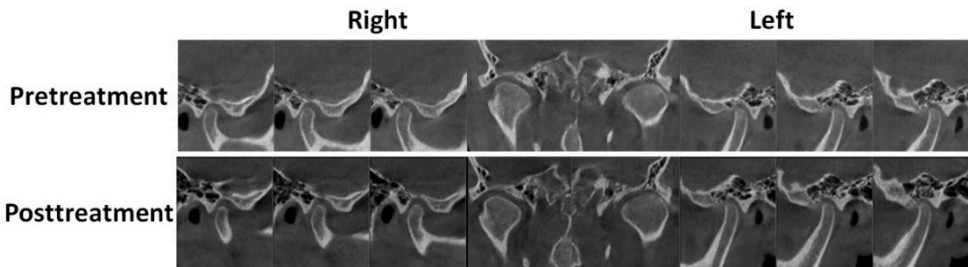

**Figure 16.** Sagittal and coronal CBCT images on the **right** and **left** condyles at pretreatment and posttreatment.

In this patient, proper occlusal interdigitation [37] and a retention protocol [34,35] were established, which were important factors in maintaining the camouflage treatment results. However, the patient showed some overbite, overjet, and dental midline relapse. Therefore, a more active retention protocol with skeletal anchorage, intra- and intermaxillary elastomers, and myofunctional therapy should be necessary [38,39].

### 4. Conclusions

Non-surgical orthodontic camouflage treatment of a skeletal Class III patient with open bite and asymmetry successfully achieved acceptable skeletal and dental changes and smile esthetics with the proper biomechanical application of multiple miniscrews and intermaxillary elastics. The treatment results were acceptable within the limitation of the camouflage treatment and were stable thanks to a customized retention protocol.

**Author Contributions:** Conceptualization, Y.-J.S., J.H.P. and J.-M.C.; writing—original draft preparation, Y.-J.S. and J.H.P.; writing—review and editing, N.-Y.C. and J.-M.C.; treating the case, Y.-J.S. and J.-M.C. All authors have read and agreed to the published version of the manuscript.

**Funding:** This research received no external funding.

**Informed Consent Statement:** Written informed consent has been obtained from the patient to publish this paper.

**Acknowledgments:** This paper was supported by Wonkwang University in 2023.

**Conflicts of Interest:** The authors declare no conflict of interest.

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
