# Peer review of "Non-Surgical Camouflage Treatment of a Skeletal Class III Patient with Anterior Open Bite and Asymmetry Using Orthodontic Miniscrews and Intermaxillary Elastics"

_applsci, doi:10.3390/app13074535_

Round 1
Reviewer 1 Report
It is a manuscript based on a case report, well-structured with good follow-up and sequence.
1. In the introduction (page 2 line 53) it mentions that the key point before starting class 3 camouflage is “the three-dimensional position of the maxillary incisors”; but the length of the roots and the quality of the alveolar bone must also be considered. Imaging (Figure 3) the upper and lower anterior teeth look in good condition, it would be good to mention it, in the CBCT description.
2. In table 1, to improve the sequence understanding, it would be good to place the correspondence times of each T.
3. It is advisable to present the cephalometric measurements at 22 months to estimate the change in relapse.
4. It is advisable to place the imaging finding presented in Figure 16 in the treatment results section.
Author Response
Response to Reviewer 1 Comments
Point 1: In the introduction (page 2 line 53) it mentions that the key point before starting class 3 camouflage is “the three-dimensional position of the maxillary incisors”; but the length of the roots and the quality of the alveolar bone must also be considered. Imaging (Figure 3) the upper and lower anterior teeth look in good condition, it would be good to mention it, in the CBCT description.
Response 1: Of course, the length of the roots and the quality of the alveolar bone must be considered for all orthodontic tooth movement. But, in this paragraph, the authors tried to focus on facial esthetics and dental occlusion in determining the 3-dimensional position of the maxillary incisors. It is because most Class III patients usually show a deficient exposure, dental midline deviation, and backward position of the maxillary incisors. In figure 3, we can see that the vertical, horizontal, and transverse positions of the maxillary incisors are abnormal in the pretreatment cephalometric radiographs. The authors added a CBCT description according to the reviewer’s comments.
“The distance between the maxillary central incisal tip and stomion was deficient, causing no incisor display when smiling. The 3D coordinates (x, y, z) were constructed us-ing nasion (N) as the reference point (0, 0, 0) (Figure 4, Table 2).”
Point 2: In table 1, to improve the sequence understanding, it would be good to place the correspondence times of each T.
Response 2: The authors added the correspondence times in Table 1 according to the reviewer’s comment. The treatment period is also described in the footnote of Table 1.
Point 3: It is advisable to present the cephalometric measurements at 22 months to estimate the change in relapse.
Response 3: The cephalometric measurements at 22 months posttreatment are described in Table 1. But the authors could not take CBCT images at 22 months posttreatment to reduce radiation exposure.
Point 4: It is advisable to place the imaging finding presented in Figure 16 in the treatment results section.
Response 4: The paragraph (page 12, lines 307-312) describes the author’s opinion, not treatment results. So it might be better to be described it in the discussion section.

Reviewer 2 Report
Authors stated that “The ideal treatment plan was a combination of surgical and orthodontic treatment” to correct the skeletal Class III asymmetry and open bite. Since the patient and his parents refused, alternative treatment was done. Authors have any examples/experience of such surgical procedures at this clinic? Did Authors find any treatment difference due to non-surgical procedures? Mention advantages and disadvantages over the non-surgical procedures.
Lane 135: Authors stated, He understood the limitations of orthodontic camouflage treatment without orthognathic surgery. What are all the limitations that were explained to the patient? Elaborate them in detail.
Author Response
Response to Reviewer 2 Comments
Point 1: Authors stated that “The ideal treatment plan was a combination of surgical and orthodontic treatment” to correct the skeletal Class III asymmetry and open bite. Since the patient and his parents refused, alternative treatment was done. Authors have any examples/experience of such surgical procedures at this clinic? Did Authors find any treatment difference due to non-surgical procedures? Mention advantages and disadvantages over the non-surgical procedures.
Response 1: The authors added the advantages and disadvantages over the non-surgical procedures.
“The authors fully explained the advantages (less aggressive procedure, low cost, acceptable treatment results without orthognathic surgery) and disadvantages (high relapse tendency, the possibility of miniscrew failure, compromised treatment results) of this non-surgical procedure to the patient. The patient chose this treatment option.”
Point 2: Lane 135: Authors stated, He understood the limitations of orthodontic camouflage treatment without orthognathic surgery. What are all the limitations that were explained to the patient? Elaborate them in detail.
Response 2: The authors added the limitations over the non-surgical procedures.
“He understood the limitations (possibility of relapse, compromise of facial, skeletal, and dental correction, the difficulty of improving facial and skeletal asymmetry) of orthodontic camouflage treatment without orthognathic surgery.”
